# Imp is required for timely exit from quiescence in *Drosophila* type II neuroblasts

Jordan A. Munroe[1], Mubarak H. Syed[2]*, Chris Q. Doe[1]*

1 Institute of Neuroscience, Howard Hughes Medical Institute, Univ. of Oregon, Eugene, OR, United States of America, 2 Department of Biology, Univ. of New Mexico, Albuquerque, NM, United States of America

* cdoe@uoregon.edu (CQD); FlyGuy@unm.edu (MHS)

**Data Availability Statement:** All relevant data are within the paper and Figs 1–4.

**Funding:** NSF CAREER award IOS-2047020 Mubarak Syed Howard Hughes Medical Institute None Chris Doe The funders had no role in study

## Abstract

Stem cells must balance proliferation and quiescence, with excess proliferation favoring tumor formation, and premature quiescence preventing proper organogenesis. *Drosophila* brain neuroblasts are a model for investigating neural stem cell entry and exit from quiescence. Neuroblasts begin proliferating during embryogenesis, enter quiescence prior to larval hatching, and resume proliferation 12-30h after larval hatching. Here we focus on the mechanism used to exit quiescence, focusing on "type II" neuroblasts. There are 16 type II neuroblasts in the brain, and they undergo the same cycle of embryonic proliferation, quiescence, and proliferation as do most other brain neuroblasts. We focus on type II neuroblasts due to their similar lineage as outer radial glia in primates (both have extended lineages with intermediate neural progenitors), and because of the availability of specific markers for type II neuroblasts and their progeny. Here we characterize the role of Insulin-like growth factor II mRNA-binding protein (Imp) in type II neuroblast proliferation and quiescence. Imp has previously been shown to promote proliferation in type II neuroblasts, in part by acting antagonistically to another RNA-binding protein called Syncrip (Syp). Here we show that reducing Imp levels delays exit from quiescence in type II neuroblasts, acting independently of Syp, with Syp levels remaining low in both quiescent and newly proliferating type II neuroblasts. We conclude that Imp promotes exit from quiescence, a function closely related to its known role in promoting neuroblast proliferation.

## Introduction

The generation of neuronal diversity is essential for proper brain assembly and function. This is particularly true for the primate cortex, which derives from a specialized neural stem cell called outer radial glia (oRG). These stem cells are thought to have driven cortical expansion and diversity during evolution [1–3], but how they regulate their proliferation remains incompletely understood.

One way to help understand oRG lineages is to use model organisms that contain neural stem cells with lineages similar to oRGs, which can be used to generate testable hypotheses for investigating primate oRG lineages. In *Drosophila*, there is a small pool of 16 neural stem cells in the brain (eight stem cells per brain lobe), called type II neuroblasts (TIINBs), that undergo a lineage similar to primate oRGs to generate neurons [4–6] (Fig 1A). In primates these oRGs

design, data collection and interpretation, or the decision to submit the work for publication.

**Competing interests:** The authors have declared that no competing interests exist.

generate neurons of the cortex; in *Drosophila* the TIINBs generate neurons of the adult central complex (CX), a region important for navigation, sleep, and sensorimotor integration [7]. Like oRGs, TIINBs undergo repeated asymmetric divisions to produce a series of transit amplifying cells called Intermediate Neural Progenitor (INPs), which themselves undergo a more limited division pattern to generate a series of ganglion mother cells (GMCs) which undergo a single terminal division to produce pairs of neurons and/or glia (Fig 1A, left) [4–6].

Neuronal diversity is generated at each step in the TIINB lineage. TIINBs change gene expression over time as they generate distinct INPs, with some genes limited to early lineage expression such as insulin-like growth factor II mRNA-binding protein (Imp), Chinmo, and Lin-28; other genes are only expressed late in the lineage such as the RNA-binding protein Syncrip (Syp), Broad, and E93 [8, 9]. These genes are called candidate temporal transcription factors (TTFs) or temporal identity factors due to their potential role in specifying different neuronal fates based on their time of birth. Subsequently, each individual INP undergoes a TTF cascade to generate molecularly distinct GMCs [10–12]. Thus, the TIINBs appear to be an excellent model for understanding oRG lineages in primates.

Another important aspect of TIINB lineages is how their pattern of proliferation is regulated to generate large populations of neurons without tumorigenesis. TIINBs begin their lineage in the embryonic brain, followed by a period of quiescence at the transition from embryo to first larval instar (L1), and then proliferation resumes between 12–30 hours after larval hatching [13, 14]; subsequently all times refer to hours after larval hatching. This is similar to the pattern of proliferation-quiescence-proliferation exhibited by most other embryonic larval neuroblast lineages [15, 16]. Previous work has shown that neuroblast quiescence is achieved through the accumulation of nuclear Prospero (Pros) [16, 17], and upon exit from quiescence each TIINB will generate ~60 INPs that produce hundreds of neurons and glia throughout larval development [4–6, 18–20]. Previous work has shown that Syp recruits the mediator complex and Pros to drive the mushroom body (MB) NBs into decommissioning [21]. This terminal exit from the cell cycle is also driven by the loss of proliferation and differentiation due to low Imp expression [21, 22]. High Imp expression in early larval life promotes neuroblast proliferation via the stabilization of *myc* and *chinmo* RNAs as well as inhibition of the mediator complex [9, 21, 23]. This makes Imp an attractive candidate for studying how TIINBs initiate exit from quiescence. Here we focus on the role of Imp in regulating neuroblast proliferation in TIINB lineages, where we identify a novel role for Imp in promoting TIINB exit from quiescence.

## Results

### Type II neuroblasts exhibit a high-to-low Imp protein gradient overtime

Previous work has shown that Imp forms a high-to-low RNA and protein gradient in all assayed neuroblast populations [23], but at just a few timepoints. Here we used *Pointed-gal4* (*pnt-gal4*), which is expressed in all TIINBs, crossed to *UAS-GFP* to identify TIINBs and co-stained for Imp at 12h intervals throughout larval stages, from 24h to 96h after larval hatching; note that all times subsequently refer to hours after larval hatching (Fig 1B). We found that Imp protein forms a gradient from high to low over the first 60h of larval life, becoming virtually undetectable from 72-96h (Fig 1B–1F). We conclude that Imp levels drop continuously in TIINBs during larval life.

### ImpRNAi and Imp overexpression have opposing effects on the timing of the Imp protein gradient in type II neuroblasts

To alter the Imp protein gradient, we performed Imp RNAi in TIINBs. We used *pnt-gal4 UAS-impRNAi* to reduce Imp protein levels specifically in TIINB lineages. We found that Imp

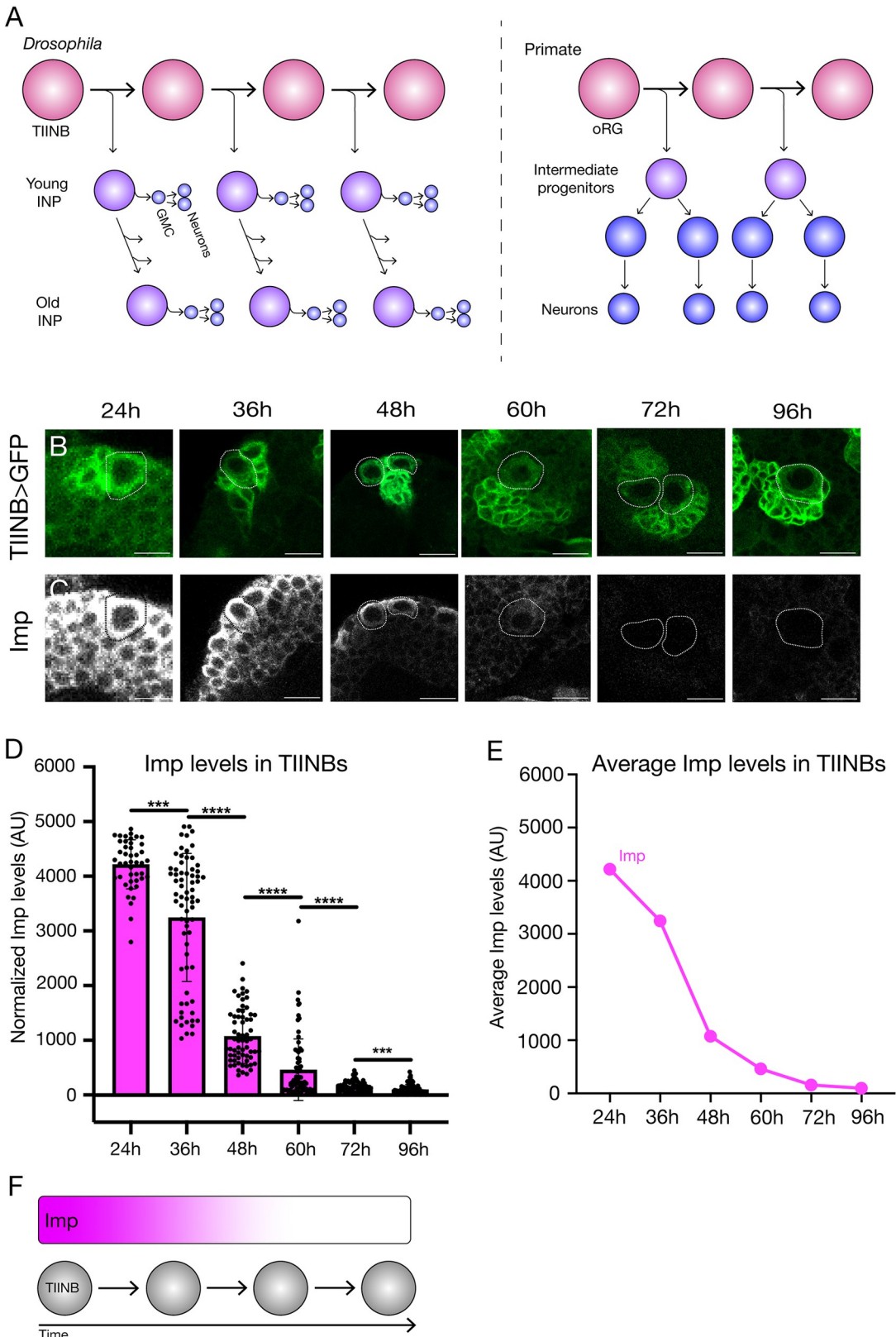

**Fig 1. Quantification of the Imp gradient in type II neuroblasts.** (A) Type II neuroblast lineage (left) [4–6] and outer radial glial lineage (right), adapted from [26]. (B,C) Imp protein forms a high-to-low gradient in type II neuroblasts during larval life

(hours are time after larval hatching in this and following figures). Type II neuroblasts are identified by expression of *pnt-gal4 UAS-GFP*. Scale bar, 20 μm. (D,E) Quantification of Imp protein levels (see methods for details) for all n's (D) or for the average levels (E). n = 5 brains, each data point is one type II neuroblast. (F) Summary.

RNAi in TIINBs significantly reduced Imp protein levels, although an Imp protein gradient persisted, effectively shifting the Imp gradient to earlier times in development (Fig 2A and 2C–2E). In contrast, overexpression of Imp within TIINB lineages results in higher levels of Imp, without abolishing its gradient, effectively shifting the Imp gradient to later times in development (Fig 2B–2E). We conclude that Imp RNAi or Imp overexpression reduces or increases Imp protein levels, respectively, and thus they are effective tools for manipulating Imp protein levels in TIINBs.

### *pnt-gal4 UAS-GFP* can be used to selectively label proliferating type II neuroblasts

Imp has been shown to promote neuroblast proliferation, and the decline in Imp levels in late larva contributes to termination of neuroblast proliferation [21, 22]. Here we asked a related question: does reduction in Imp levels in TIINB delay exit from quiescence? Proliferating versus quiescent TIINBs can be distinguished by expression of *pnt-Ga4 UAS-GFP*, Deadpan (Dpn) and Cyclin E (CycE): proliferative neuroblasts in interphase are GFP+Dpn+CycE + whereas quiescent neuroblasts are GFP-Dpn+CycE- [15, 16]. We found that *pnt-Gal4 UAS-GFP* was only expressed by proliferating TIINBs (Fig 3A; quantified in 3C), and no quiescent neuroblasts expressed *pnt-Gal4 UAS-GFP* (Fig 3B; quantified in 3C). This allowed us to quantify how many of the 16 TIINBs were proliferating, and infer the remainder were quiescent (see below). We conclude that *pnt-gal4 UAS-GFP* can be used to identify proliferating TIINBs (Fig 3D).

### Imp is required for timely exit from quiescence in type II neuroblasts

High Imp expression early in larval development promotes neuroblast proliferation, while late, low Imp expression leads to neuroblast decommissioning [21, 22]. We wanted to know if high Imp expression early in larval life promoted TIINB exit from quiescence. To answer this question, we decreased Imp levels specifically in TIINB lineages and quantified the number of proliferating TIINBs at intervals from 24h to 96h. We used *pnt-gal4 UAS-GFP* to identify proliferating TIINBs, *UAS-ImpRNAi* (to reduce Imp levels), and Dpn to mark all neuroblasts (proliferating or quiescent). In wild type, at 24h ~8 of the 16 TIINBs are *pnt-Gal4 UAS-GFP* + and thus have exited quiescence, with the remainder still in quiescence. By 36h, all 16 TIINBs have exited quiescence and are proliferative (Fig 4A and 4B). In contrast, following Imp RNAi, only ~2 TIINBs have exited quiescence at 24h, and it takes until 72h for all 16 TIINBs to exit quiescence and become proliferative (Fig 4A and 4B). We also wanted to see if Imp RNAi delayed exit from quiescence in specific TIINB lineages–e.g. the pair of lateral DL neuroblasts or dorsomedial DM neuroblasts–but each class had an indistinguishable time of exit from quiescence. We conclude that Imp promotes exit from quiescence in TIINBs.

To determine if higher levels of Imp could drive precocious exit from quiescence, we used *pnt-gal4* to drive *UAS-Imp* specifically in TIINB lineages. This manipulation results in significantly more Imp protein in TIINBs (Fig 2), but overexpression of Imp does not induce precocious exit from quiescence in TIINBs (Fig 4C and 4D). We conclude that Imp is necessary but not sufficient to drive TIINB exit from quiescence.

Because Imp promotes exit from quiescence, we asked whether quiescent TIINBs have low Imp and proliferating TIINBs have high Imp levels. Interestingly, we observed comparable

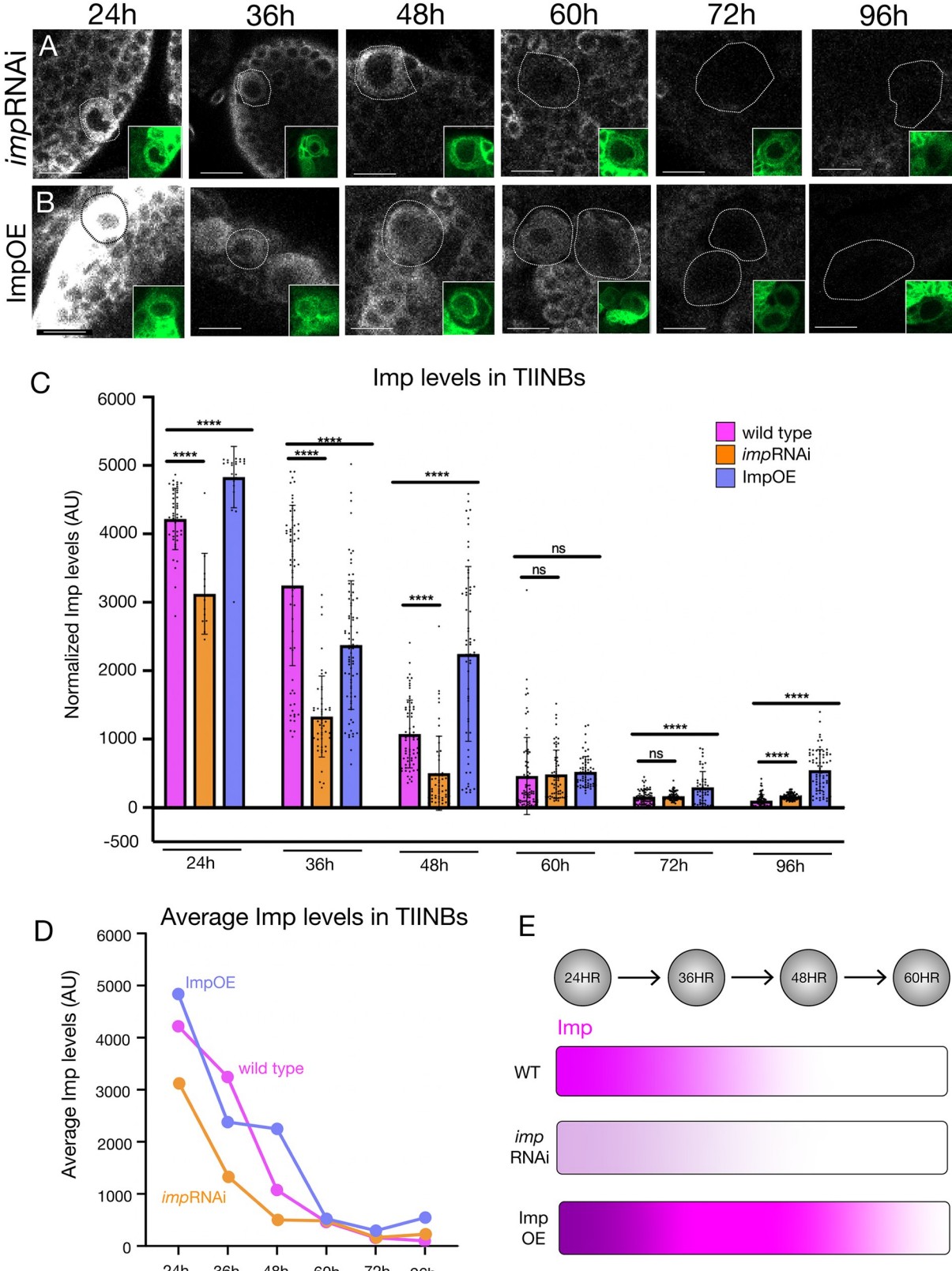

**Fig 2. Imp RNAi and Imp overexpression result in reduced or increased Imp protein levels.** Wild type Imp levels are shown in Fig 1. (A) Imp RNAi within type II neuroblasts (inset: *pnt-gal4 UAS-GFP*) leads to lower Imp levels without disrupting the protein gradient; quantified in

C. Scale bar, 20 μm. (C) Imp overexpression within type II neuroblasts (inset: *pnt-gal4 UAS-GFP*) leads to higher Imp levels without disrupting the protein gradient; quantified in C. Scale bar, 20 μm. (C, D) Quantification of Imp protein levels in type II neuroblasts in wild type, Imp RNAi, and Imp overexpression. (C) Histogram showing all n's; (D) graph showing average values. n = 5 brains, each data point is one type II neuroblast. (E) Summary.

levels of Imp in proliferating TIINBs (Fig 4E, first column; quantified in 4F) and quiescent TIINBs (Fig 4E, second column; quantified in 4F). Because Imp and Syp can cross-repress each other [23], we assayed Syp levels in proliferating and quiescent TIINBs. As expected, we found Syp to be expressed at lower levels than Imp in both proliferating and quiescent TIINBs (Fig 4E, third and fourth columns; quantified in 4F). Previous work has shown little to no Syp expression in early TIINBs; the very low levels of Syp seen here may be due to more sensitive acquisition methods than used previously [8]. Interestingly, Syp levels in quiescent TIINBs were slightly higher than Syp levels in proliferative TIINBs (Fig 4F), showing a correlation between higher Syp levels and neuroblast quiescence. We conclude that Imp is expressed in quiescent neuroblasts and is necessary but not sufficient for timely exit from quiescence (Fig 4G).

## Discussion

It is well documented in previous studies that Imp is expressed in a temporal gradient in many central brain neuroblasts [8, 9, 22, 23]. In this study we have confirmed the Imp gradient in TIINBs from 24h – 96h and have quantified Imp levels in wild type as well as after Imp RNAi knockdown or Imp overexpression. While both knockdown and overexpression show significant changes in Imp levels, the Imp gradient is maintained throughout larval life in all cases. Interestingly, at 36h Imp overexpression levels are lower than WT control levels, but only at this timepoint. This suggests a post-transcriptional 'homeostatic' mechanism that reduces Imp levels when they are experimentally increased. A possible explanation for this is Imp targeting by microRNA *let-7*. *let-7* targets Imp in *Drosophila* male testis [24] and is present in MB NBs where it targets the temporal transcription factor Chinmo, which is also present in TIINBs [25]. Thus, *let-7* may regulate Imp in TIINBs and should be explored in future work.

At 24h wild type larval brains show ~8–10 TIINBs active, and all 16 TIINBs (8 neuroblasts per brain lobe) are active and proliferating by 36h. Imp knockdown results in only ~2–4 TIINBs at 24h and all 16 TIINBs are not proliferating until 72h. This late exit from quiescence shows that Imp is necessary for timely exit from quiescence. Previous studies have shown that high levels of Imp in TIINBs are required to maintain large neuroblast size and proliferative activity through the stabilization of *myc* RNA [22], and overexpression of Imp in neuroblasts can extend proliferation [21, 22]. Our results add to these findings by showing that Imp is required for TIINB timely exit from quiescence. Additionally, Imp knock down in TIINBs promotes early exit from cell cycle at the end of larval life [21]. Imp overexpression in TIINBs did not change the rate at which TIINBs exit from quiescence. Thus, Imp is necessary but not sufficient for exit from quiescence. These findings suggest that a minimum level of Imp is required for the exit from quiescence. A potential mechanism for this would be a negative feedback loop driven by over-expression of Imp, which could lead to over-proliferation if not regulated. Again, a candidate factor for regulation of Imp levels as TIINBs exit quiescence is *let-7*.

It is interesting that Pnt-gal4 is not expressed in quiescent neuroblasts, yet is able to drive *UAS-ImpRNAi* at sufficient levels to maintain quiescence. Type II NBs are proliferative in the embryo, then go quiescent, and normally resume proliferation in 12-30h old larvae. We propose that *pnt-ga4* is expressed in the embryo type II neuroblasts where it drives *UAS-ImpRNAi*

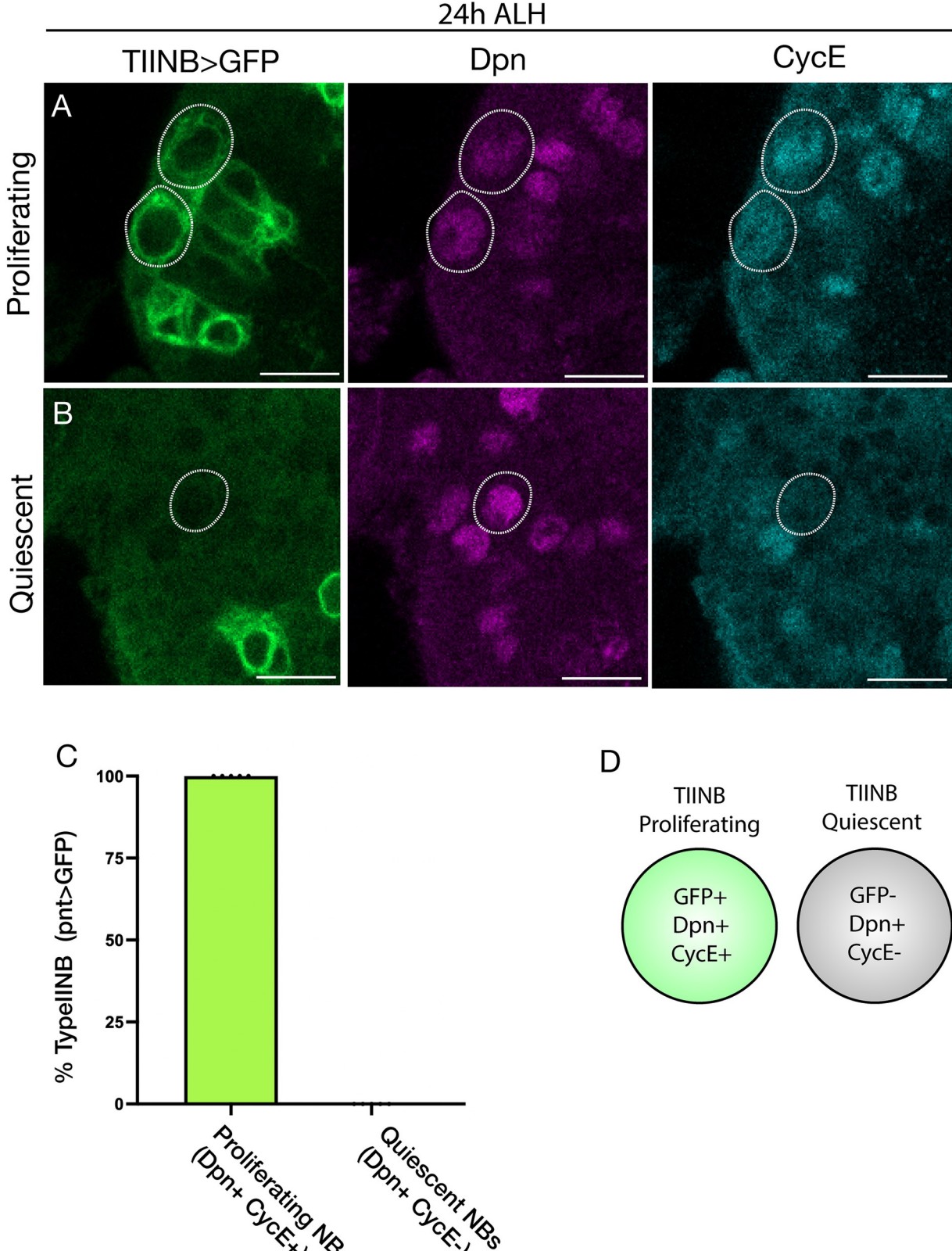

**Fig 3. Pointed-gal4 UAS-GFP+ TIINBs have exited quiescence and are proliferative.** (A) Type II neuroblasts are circled and identified by *pnt-gal4 UAS-GFP* (green), Dpn (magenta), and reconfirmed as proliferative by CycE (cyan) at 24h. Scale bar is 5 μm. (B) *pnt-gal4 UAS-GFP*

(green) and CycE (cyan) are not expressed in quiescent type II neuroblasts, but Dpn (magenta) is still present. Quiescent cells (circled) are identified based on their position in the brain. Scale bar is 5 μm. (C) Histogram of cells that are Dpn+. One hundred percent of type II neuroblasts that are positive for GFP (*pnt-gal4 UAS-GFP*) are Dpn+ and CycE+, while 0% of cells that are GFP-, Dpn+, CycE-. n = 5 brains, each data point represents one brain. (D) Summary.

which persists into larval stages due to perdurance of Gal4 and Imp RNAi, thus extending quiescence. As Imp RNAi levels begin to rise (due to lack of *pnt-gal4 UAS-ImpRNAi* expression) the neuroblasts resume proliferation. We see no evidence for a second wave of quiescence due to re-expression of *pnt-gal4*.

We quantified Imp levels in both quiescent and proliferative TIINBs to see how they varied and saw no change. We also wanted to compare Syp levels to Imp levels in quiescent and proliferative TIINBs. Syp is required for the entrance into quiescence and decommissioning [21], but it was unknown what Syp levels are in TIINBs nearing the end of quiescence early in larval life. We compared Syp levels in proliferating TIINBs to quiescent TIINBs but found that Syp levels were significantly lower than Imp levels, consistent with their cross-repressive regulation. Interestingly, Syp levels in quiescent TIINBs were higher than Syp levels in proliferative TIINBs, showing a correlation between high Syp levels and neuroblast quiescence, and consistent with earlier work showing Syp is required to elevate levels of nuclear Prospero and initiate neuroblast decommissioning [21].

## Materials and methods

### Fly stocks

; *UAS-myr::GFP; pointed-Gal4*
   ; *UAS-myr::GFP; pointed-Gal4*
   ;; *UAS-ImpRNAi*
   *UAS-Imp; Sco/Cyo*
   ; *UAS-myr::GFP; pointedGal4*

### Antibodies and immunostaining

We used the following antibodies: chicken GFP (Abcam, Eugene OR 1:1000), rabbit Imp (McDonald lab, UT Austin, 1:1000), rabbit Syp (Desplan lab, NYU, 1:1000), rat Deadpan (Dpn; Abcam, Eugene OR 1:20), rabbit Cyclin E (CycE; Santa Cruz Biotech, #C1209, 1:500), guinea pig Asense (Wang lab, Duke, 1:500), and secondary antibodies were from Thermofisher, Eugene OR used at manufacturer's recommendation. All larvae were raised at 25˚C and dissected in Hemolymph Like buffer 3.1 (HL3.1) (NaCl 70mM, KCl 5mM, CaCl$_2$ 1.5mM, MgCl$_2$ 4mM, sucrose 115mM, HEPES 5mM, NaHCO$_3$ 10mM, and Trehalose 5mM in double distilled water). Larvae were grown to specified time points, dissected, mounted on poly-D-lysine coated slips (Neuvitro, Camas, WA), and incubated for 30 minutes in 4% paraformaldehyde solution in Phosphate Buffered Saline (PBS) with 1% Triton-X (1% PBS-T) at room temperature. Larval brains were washed twice with 0.5% PBS-T and incubated for 1–7 days at 4˚C in a blocking solution of 1% goat serum (Jackson ImmunoResearch, West Grove, PA), 1% donkey serum (Jackson ImmunoResearch, West Grove, PA), 2% dimethyl sulfoxide in organosulfur (DMSO), and 0.003% bovine serum albumin (BSA) (Fisher BioReagents, Fair Lawn, NJ Lot #196941). Larval brains were incubated overnight at 4˚C in a solution of primary antibodies in 0.5% PBS-T. Larval brains were washed for at least 60 minutes in 0.5% PBS-T at room temperature, and then incubated overnight at 4˚C in a solution of secondary antibodies in 0.5% PBS-T. Brains were washed in 0.5% PBS-T for at least 60 minutes at room temperature.

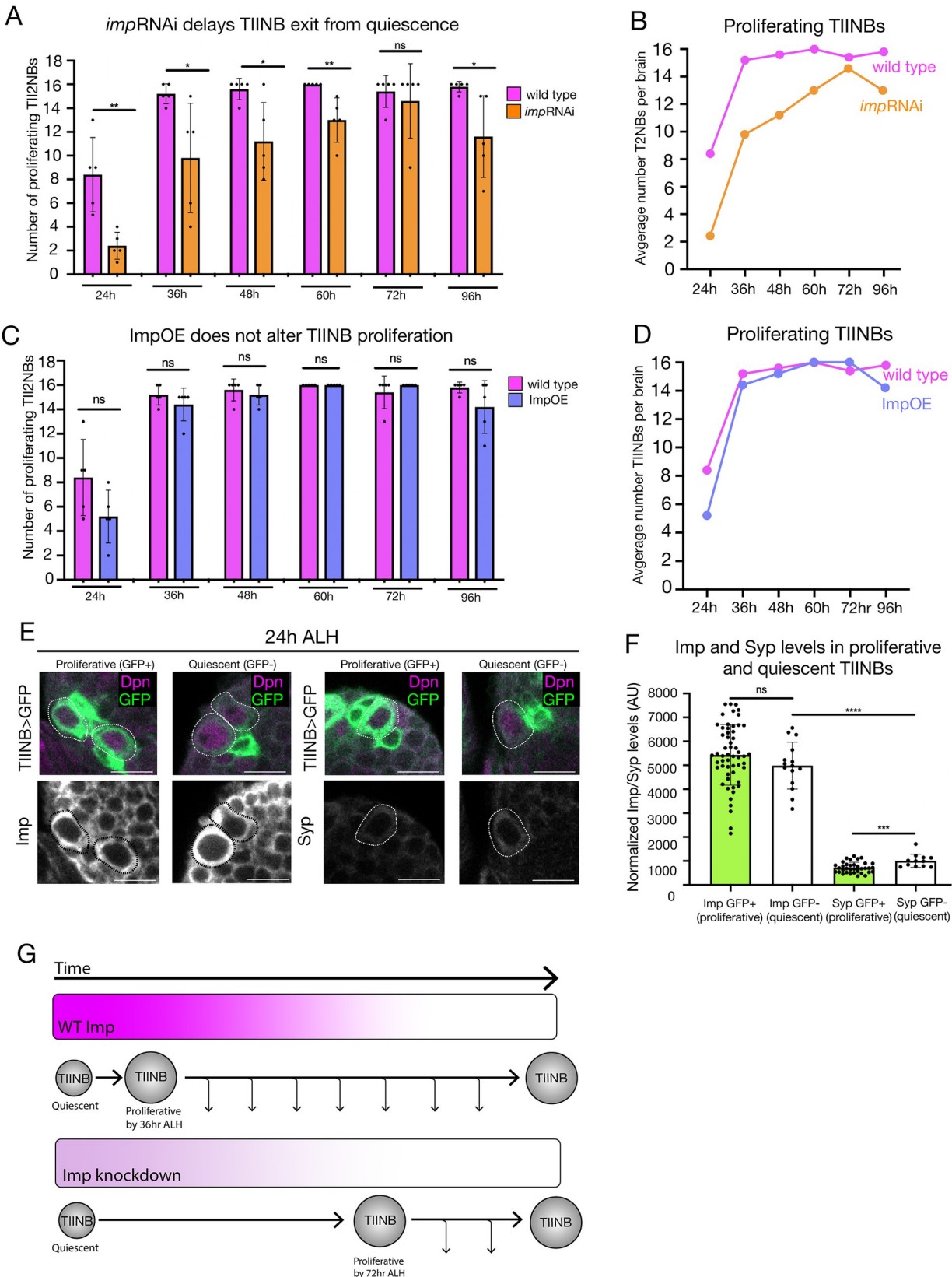

**Fig 4. Imp is required for timely exit from quiescence in type II neuroblasts.** (A,B) Quantification of proliferating type II neuroblast numbers (expressing *pnt-gal4 UAS-GFP*) over larval life in wild type and Imp RNAi. Note that there is a maximum of 16 type II neuroblasts per brain. In wild type, all neuroblasts have exited quiescence/resumed proliferating by 36h as shown by *pnt-gal4 UAS-GFP* expression. Imp RNAi delays exit from quiescence and the full complement of 16 proliferating type II neuroblasts is not achieved until 72h as shown by *pnt-gal4 UAS-GFP* expression. n = 5 brains, each data point represents one brain. (C,D) Quantification of proliferating type II neuroblast numbers (*pnt-gal4 UAS-GFP*+) across larval development for wild type and Imp overexpression. There is no difference in exit from quiescence between wild type and Imp overexpression genotypes. (E) Imp levels are the same in quiescent and proliferating type II neuroblasts, while Syp levels are lower in proliferating type II neuroblasts. Proliferating type II neuroblasts (circled; first and third columns) are identified by expression of *pointed-gal4 UAS-GFP* (green), Dpn (magenta), and lack of Asense (not shown). Quiescent type II neuroblasts do not express *pointed-gal4 UAS-GFP* (green) but can be identified as Dpn+ (magenta) and lack of Asense. n = 5 brains, each data point represents one brain. (F) Quantification of Imp and Syp levels in quiescent and proliferating type II neuroblasts at 24h. n = 5 brains, each data point is one type II neuroblast. (G) Summary.

Brains were dehydrated by going through a series of 10-minute washes in 30%, 50%, 70%, and 90% EtOH, and two rounds of 10 minutes in 100% EtOH and two rounds of 10 minutes in xylene (MP Biomedicals, LLC, Saolon, OH, Lot# S0170), then mounted in dibutyl phthalate in xylene (DPX; Sigma-Aldrich, cat. no. 06522). Brains sat in DPX for at least 48 hours at 4°C or 72 hours (48 hours at room temperature and 24 hours at 4°C) before imaging.

## Imaging and statistical analysis

All Imp data were collected with identical confocal settings; all Syp data were collected with identical confocal settings. Fluorescent images were collected on Zeiss LSM 800. TIINBs were counted using the cell counter plugin in FIJI (https://imagej.net/software/fiji/). Imp pixel density in each TIINB was calculated in FIJI. In FIJI, TIINBs were manually selected in a 2D plane at the largest cross section of the TIINB with the polygon lasso tool, and the area and Raw Integrated Density (RID) was measured. The nucleus of each TIINB went through the same analysis steps. Imp is cytoplasmic and measuring fluorescence in the nucleus functioned as background subtraction. Imp levels were normalized to cell area using the equation: (Cell Body$^{RID}$–Nucleus$^{RID}$) / (Cell Body$^{Area}$–Nucleus$^{Area}$). Two-tailed student t-tests were used to compare two sets of data. *$p < 0.05$; **$p < 0.01$; ***$p < 0.001$; ****$p < 0.0001$. All graphs and statistical analysis were done in Prism (GraphPad Software, San Diego, CA). Note that we were unable to quantify Imp fluorescence in quiescent TIINBs in Imp RNAi flies because quiescent TIINBs cannot be distinguished from quiescent Type I neuroblasts.

## Figure production

Images for figures were taken in FIJI. Figures were assembled in Adobe Illustrator (Adobe, San Jose, CA). Any changes in brightness or contrast were applied to the entire image.

## Acknowledgments

We thank Noah Dillon and Gonzalo Morales Chaya for comments on the manuscript, and Adam Fries for help with developing a fluorescent analysis method.

## Author Contributions

**Conceptualization:** Jordan A. Munroe, Chris Q. Doe.

**Data curation:** Jordan A. Munroe, Chris Q. Doe.

**Formal analysis:** Jordan A. Munroe, Mubarak H. Syed.

**Funding acquisition:** Jordan A. Munroe, Mubarak H. Syed, Chris Q. Doe.

**Investigation:** Jordan A. Munroe.

**Methodology:** Jordan A. Munroe.

**Project administration:** Mubarak H. Syed, Chris Q. Doe.

**Supervision:** Mubarak H. Syed, Chris Q. Doe.

**Writing – original draft:** Jordan A. Munroe, Chris Q. Doe.

**Writing – review & editing:** Jordan A. Munroe, Mubarak H. Syed, Chris Q. Doe.

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
