## [Decision Letter · Decision Letter 0]

23 Aug 2022

PONE-D-22-19798Imp is required for timely exit from quiescence in Drosophila type II neuroblastsPLOS ONE

Dear Dr. Doe,

Thank you for submitting your manuscript to PLOS ONE. After careful consideration, we feel that it has merit but does not fully meet PLOS ONE’s publication criteria as it currently stands. Therefore, we invite you to submit a revised version of the manuscript that addresses the points raised during the review process.

We look forward to receiving your revised manuscript.

Kind regards,

Hongyan Wang, Ph.D.

Academic Editor

PLOS ONE

Journal Requirements:When submitting your revision, we need you to address these additional requirements. 1. Please ensure that your manuscript meets PLOS ONE's style requirements, including those for file naming. The PLOS ONE style templates can be found at https://journals.plos.org/plosone/s/file?id=wjVg/PLOSOne_formatting_sample_main_body.pdf and https://journals.plos.org/plosone/s/file?id=ba62/PLOSOne_formatting_sample_title_authors_affiliations.pdf 2. We note that the grant information you provided in the ‘Funding Information’ and ‘Financial Disclosure’ sections do not match.  When you resubmit, please ensure that you provide the correct grant numbers for the awards you received for your study in the ‘Funding Information’ section. 3. Thank you for stating the following in the Acknowledgments Section of your manuscript:   "Funder
Grant reference number
AuthorNIH
T32HD07348
Jordan MunroeNSF CAREER award 
IOS-2047020
Mubarak SyedHoward Hughes Medical Institute
None 
Chris Doe" We note that you have provided funding information that is not currently declared in your Funding Statement. However, funding information should not appear in the Acknowledgments section or other areas of your manuscript. We will only publish funding information present in the Funding Statement section of the online submission form. Please remove any funding-related text from the manuscript and let us know how you would like to update your Funding Statement. Currently, your Funding Statement reads as follows:  "CD - HHMI InvestigatorJM - NIH Developmental Biology Training grant" Please include your amended statements within your cover letter; we will change the online submission form on your behalf. 4. Thank you for stating the following in your Competing Interests section:   "NO authors have competing interests" Please complete your Competing Interests on the online submission form to state any Competing Interests. If you have no competing interests, please state "The authors have declared that no competing interests exist.", as detailed online in our guide for authors at http://journals.plos.org/plosone/s/submit-now  This information should be included in your cover letter; we will change the online submission form on your behalf. 5. In your Data Availability statement, you have not specified where the minimal data set underlying the results described in your manuscript can be found. PLOS defines a study's minimal data set as the underlying data used to reach the conclusions drawn in the manuscript and any additional data required to replicate the reported study findings in their entirety. All PLOS journals require that the minimal data set be made fully available. For more information about our data policy, please see http://journals.plos.org/plosone/s/data-availability. Upon re-submitting your revised manuscript, please upload your study’s minimal underlying data set as either Supporting Information files or to a stable, public repository and include the relevant URLs, DOIs, or accession numbers within your revised cover letter. For a list of acceptable repositories, please see http://journals.plos.org/plosone/s/data-availability#loc-recommended-repositories. Any potentially identifying patient information must be fully anonymized. Important: If there are ethical or legal restrictions to sharing your data publicly, please explain these restrictions in detail. Please see our guidelines for more information on what we consider unacceptable restrictions to publicly sharing data: http://journals.plos.org/plosone/s/data-availability#loc-unacceptable-data-access-restrictions. Note that it is not acceptable for the authors to be the sole named individuals responsible for ensuring data access. We will update your Data Availability statement to reflect the information you provide in your cover letter. 6. We note that you have included the phrase “data not shown” in your manuscript. Unfortunately, this does not meet our data sharing requirements. PLOS does not permit references to inaccessible data. We require that authors provide all relevant data within the paper, Supporting Information files, or in an acceptable, public repository. Please add a citation to support this phrase or upload the data that corresponds with these findings to a stable repository (such as Figshare or Dryad) and provide and URLs, DOIs, or accession numbers that may be used to access these data. Or, if the data are not a core part of the research being presented in your study, we ask that you remove the phrase that refers to these data.7. Please review your reference list to ensure that it is complete and correct. If you have cited papers that have been retracted, please include the rationale for doing so in the manuscript text, or remove these references and replace them with relevant current references. Any changes to the reference list should be mentioned in the rebuttal letter that accompanies your revised manuscript. If you need to cite a retracted article, indicate the article’s retracted status in the References list and also include a citation and full reference for the retraction notice.

Reviewers' comments:

Reviewer's Responses to Questions

**Comments to the Author**

1. Is the manuscript technically sound, and do the data support the conclusions?

Reviewer #1: Yes

Reviewer #2: Yes

Reviewer #3: Partly

2. Has the statistical analysis been performed appropriately and rigorously? 

Reviewer #1: Yes

Reviewer #2: Yes

Reviewer #3: Yes

3. Have the authors made all data underlying the findings in their manuscript fully available?

Reviewer #1: No

Reviewer #2: Yes

Reviewer #3: No

4. Is the manuscript presented in an intelligible fashion and written in standard English?

Reviewer #1: Yes

Reviewer #2: Yes

Reviewer #3: Yes

5. Review Comments to the Author

Reviewer #1: Balancing the proliferation and quiescence is essential for the normal functions of stem cells. In this manuscript, Munroe et al investigated the role of Insulin-like growth factor II mRNA- binding protein (Imp) in proliferation and quiescence of Drosophila type II neuroblasts (TIINBs). They found Imp displayed a high-to-low temporal gradient in TIINBs and the Imp gradient can be effectively shifted to different times in development by ImpRNAi and Imp overexpression. They further suggested that pnt-Gal4 UAS-GFP could be used as a marker to identify proliferating type II neuroblasts. By quantifying the number of proliferating TIINBs, they found that Imp knockdown delayed exit from quiescence, whereas overexpression of Imp did not induce precocious exit from quiescence in TIINBs. Moreover, comparable levels of Imp were observed in quiescent and proliferating TIINBs while its antagonist protein Syp expressed at a significantly lower level.

Overall, this paper suggests a necessary but not sufficient role of Imp in mediating timely exit from quiescence in TIINBs. Experiments were carefully performed, the data presented are of high quality, and interpretations of the results are comparatively justified. I only have several minor suggestions for the authors to consider:

1. For all the Imp/syp immunostaining data, the numbers of TIINBs for statistical analysis were not provided in Methods or Figure legends.

2. Line 133,“Syp levels in quiescent TIINBs were slightly higher than Syp levels in proliferative TIINBs (Figure 4F), showing a correlation between higher Syp levels and neuroblast quiescence.”Whereas the figure legend of Fig4E claimed that “Imp and Syp levels are the same in quiescent and proliferating type II neuroblasts.”Based on the data, the legend needs to be fixed.

Reviewer #2: Munroe et al. report a role for Imp in exit from quiescence of type II neuroblasts in Drosophila. They examine and quantify Imp protein levels in Type II neuroblasts at different larval stages, something that had been missing (although much needed) in the field. They also test whether Imp regulates exit from quiescence based on high to low Imp expression and based on the known role of Imp in regulating neuroblast" decommissioning". While the data presented in this manuscript is solid and well presented, this reviewer has one point that may need to be further addressed:

Authors report that Pnt-Gal4 is only active in proliferating type II neuroblasts. By this logic, using Pnt-Gal4 should only knockdown Imp in proliferating neuroblasts. Is knockdown of Imp delaying the exit from quiescence or suppressing proliferation? or both? Is there data to suggest that Pnt-GAL4/UAS transgene expression perdures in quiescent type II neuroblasts?

Typo:

Line 66, It is the non-mushroom body neuroblasts that require low Notch signaling to be driven out of quiescence. MB neuroblasts do not enter and exit quiescence. Authors should also check the reference for this statement.

Reviewer #3: 

Overall evaluation and significance

Munroe et al., investigate the role of Imp, an important RNA-binding protein (RBP) involved in Drosophila neurogenesis, in neuroblast (NB) quiescence exit. Re-entry of NBs into cell cycle has been previously shown to be regulated by nutrition-dependent glial niche and the Notch signalling pathway, while the role of Imp in this process has not been explored. Here, the authors characterised the kinetics of Type II NB quiescence exit upon modulation of the Imp expression gradient. They observed that reducing the level of Imp, which accelerates Imp protein depletion in Type II NBs, delays exit from quiescence, while the overexpression of Imp did not lead to a noticeable effect. Taken together, the authors suggest a previously unknown role of Imp in promoting re-entry of quiescent NBs into cell cycle. The manuscript is high quality and of value and interest. It should be accepted for publication, once the authors have addressed our concerns, either by revising the text or if they would prefer, providing additional data.

Major comments

The authors first present a detailed comparison of Imp expression kinetics in Imp RNAi and Imp OE experiments. The study then aims to demonstrate that modulating Imp affects Type II NB quiescence exit. However, their choice of PntP1-GAL4 driver needs to be explained / justified, in light of the main conclusion of this work.

1. Throughout the manuscript, the authors utilise PntP1-GAL4 line to identify proliferating NBs and to drive UAS transgenes. However, in Figure 3, the authors describe that PntP1-GAL4 only becomes active in proliferating NBs and not in quiescent cells. Therefore, the Imp RNAi should be inactive in quiescent NBs, which does not support the authors' conclusion that reduced level of Imp leads to delayed quiescence exit. The authors should include more information that justifies their choice of PntP1-GAL4 over more conventionally used wor-GAL4, ase-GAL80.

2. Although the knock down of Imp in Figure 2C-D seems convincing on the population-wide scale, a key piece of data is missing that GFP-negative quiescent NBs are under PntP1-GAL4/Imp-RNAi control (Figure 4A). Can the authors provide an explanation or data that Imp levels are reduced in quiescent NBs in Imp RNAi conditions compared to the wild-type at 24-48 ALH, and also discuss why Imp RNAi is active despite the lack of GFP expression? Is it possible that other cell types (e.g. glia) might be affected by the driver?

3. Are all PntP1>GFP-positive NBs CycE-positive? The population of GFP-positive but CycE-negative cells may suggest that quiescence exit is a multi-step process where Imp plays a role in the initial step.

Minor comments

1. Figure 2A-B: Representative wild-type series of Type II NB images should be provided to match the quantification shown in Figure 2C.

2. Line 213: Please provide replicate information in all figure legends. In particular, what does the symbol represent in Figure 4A and 4C? It should be clear how many brains were quantified per biological replicate.

3. Figure 4E-F: Syed et al., 2017 (doi.org/10.7554/eLife.26287) have shown lack of Syp expression in Type II NBs at stages before 48 ALH and upon disruption of ecysone signalling. Can the authors explain the discrepancy?

4. Line 197: Immunofluorecence quantification method should be explained in more detail. Was the Raw Integrated Density calculated over 3D volume of the NBs or on select 2D planes? Why were nuclear areas and intensities removed from the analysis? Were any background subtraction method used?

5. In Figure 2C, how were the fluorescence intensity signals normalised between biological replicates and different genotypes?

6. Line 168: The role of Syp in entering the embryo-to-larval neuroblast quiescence is not yet established. The text should be revised. Perhaps the authors meant 'decomissioning and cell cycle exit'?

6. PLOS authors have the option to publish the peer review history of their article (what does this mean?). If published, this will include your full peer review and any attached files.

Reviewer #1: No

Reviewer #2: No

Reviewer #3: No

---

## [Author Response · Author response to Decision Letter 0]

26 Aug 2022

Reviewer #1: Balancing the proliferation and quiescence is essential for the normal functions of stem cells. In this manuscript, Munroe et al investigated the role of Insulin-like growth factor II mRNA- binding protein (Imp) in proliferation and quiescence of Drosophila type II neuroblasts (TIINBs). They found Imp displayed a high-to-low temporal gradient in TIINBs and the Imp gradient can be effectively shifted to different times in development by ImpRNAi and Imp overexpression. They further suggested that pnt-Gal4 UAS-GFP could be used as a marker to identify proliferating type II neuroblasts. By quantifying the number of proliferating TIINBs, they found that Imp knockdown delayed exit from quiescence, whereas overexpression of Imp did not induce precocious exit from quiescence in TIINBs. Moreover, comparable levels of Imp were observed in quiescent and proliferating TIINBs while its antagonist protein Syp expressed at a significantly lower level. Overall, this paper suggests a necessary but not sufficient role of Imp in mediating timely exit from quiescence in TIINBs. Experiments were carefully performed, the data presented are of high quality, and interpretations of the results are comparatively justified. I only have several minor suggestions for the authors to consider:

1. For all the Imp/syp immunostaining data, the numbers of TIINBs for statistical analysis were not provided in Methods or Figure legends.

Thank you for the observation, in response we have added quantification of brain number and neuroblast number to the relevant figure legends. 

2. Line 133,“Syp levels in quiescent TIINBs were slightly higher than Syp levels in proliferative TIINBs (Figure 4F), showing a correlation between higher Syp levels and neuroblast quiescence.” Whereas the figure legend of Fig4E claimed that “Imp and Syp levels are the same in quiescent and proliferating type II neuroblasts. “Based on the data, the legend needs to be fixed.

We agree, and have made the following change in the figure legend of Fig4E: "... Syp levels are lower in proliferating type II neuroblasts." 

Reviewer #2: Munroe et al. report a role for Imp in exit from quiescence of type II neuroblasts in Drosophila. They examine and quantify Imp protein levels in Type II neuroblasts at different larval stages, something that had been missing (although much needed) in the field. They also test whether Imp regulates exit from quiescence based on high to low Imp expression and based on the known role of Imp in regulating neuroblast" decommissioning". While the data presented in this manuscript is solid and well presented, this reviewer has one point that may need to be further addressed:

Authors report that Pnt-Gal4 is only active in proliferating type II neuroblasts. By this logic, using Pnt-Gal4 should only knockdown Imp in proliferating neuroblasts. Is knockdown of Imp delaying the exit from quiescence or suppressing proliferation? or both? Is there data to suggest that Pnt-GAL4/UAS transgene expression perdures in quiescent type II neuroblasts?

Great point! We've added the following text to the Discussion. "It is interesting that Pnt-gal4 is not expressed in quiescent neuroblasts, yet is able to drive UAS-ImpRNAi at sufficient levels to maintain quiescence. Type II NBs are proliferative in the embryo, then go quiescent, and normally resume proliferation in 12-30h old larvae. We propose that Pnt-ga4 is expressed in the embryo type II neuroblasts where it drives UAS-ImpRNAi which persists into larval stages due to perdurance of Gal4 and ImpRNAi, thus extending quiescence. As ImpRNAi levels begin to rise (due to lack of Pnt-Gal4 UAS-ImpRNAi expression) the neuroblasts resume proliferation. We see no evidence for a second wave of quiescence due to re-expression of Pnt-gal4."

Typo: Line 66, It is the non-mushroom body neuroblasts that require low Notch signaling to be driven out of quiescence. MB neuroblasts do not enter and exit quiescence. Authors should also check the reference for this statement.

Thanks for catching this error; we have changed the text to say "previous work has shown that Syp recruits the mediator complex and Pros to drive the MB NBs into decommissioning (21)."

Reviewer #3: 

Overall evaluation and significance

Munroe et al., investigate the role of Imp, an important RNA-binding protein (RBP) involved in Drosophila neurogenesis, in neuroblast (NB) quiescence exit. Re-entry of NBs into cell cycle has been previously shown to be regulated by nutrition-dependent glial niche and the Notch signaling pathway, while the role of Imp in this process has not been explored. Here, the authors characterised the kinetics of Type II NB quiescence exit upon modulation of the Imp expression gradient. They observed that reducing the level of Imp, which accelerates Imp protein depletion in Type II NBs, delays exit from quiescence, while the overexpression of Imp did not lead to a noticeable effect. Taken together, the authors suggest a previously unknown role of Imp in promoting re-entry of quiescent NBs into cell cycle. The manuscript is high quality and of value and interest. It should be accepted for publication, once the authors have addressed our concerns, either by revising the text or if they would prefer, providing additional data.

Major comments

The authors first present a detailed comparison of Imp expression kinetics in Imp RNAi and Imp OE experiments. The study then aims to demonstrate that modulating Imp affects Type II NB quiescence exit. However, their choice of PntP1-GAL4 driver needs to be explained / justified, in light of the main conclusion of this work.

1. Throughout the manuscript, the authors utilise PntP1-GAL4 line to identify proliferating NBs and to drive UAS transgenes. However, in Figure 3, the authors describe that PntP1-GAL4 only becomes active in proliferating NBs and not in quiescent cells. Therefore, the Imp RNAi should be inactive in quiescent NBs, which does not support the authors' conclusion that reduced level of Imp leads to delayed quiescence exit. The authors should include more information that justifies their choice of PntP1-GAL4 over more conventionally used wor-GAL4, ase-GAL80.

Great point! We've added the following text to the Discussion. "It is interesting that Pnt-gal4 is not expressed in quiescent neuroblasts, yet is able to drive UAS-ImpRNAi to maintain quiescence. Type II NBs are proliferative in the embryo, then go quiescent, and normally resume proliferation in 12-30h old larvae. We propose that Pnt-ga4 is expressed in the embryo type II neuroblasts where it drives UAS-ImpRNAi which persists into larval stages due to perdurance of ImpRNAi, thus extending quiescence. As ImpRNAi levels begin to rise (due to lack of Pnt-Gal4 UAS-ImpRNAi expression) the neuroblasts resume proliferation. We see no evidence for a second wave of quiescence due to re-expression of Pnt-gal4."

2. Although the knock down of Imp in Figure 2C-D seems convincing on the population-wide scale, a key piece of data is missing that GFP-negative quiescent NBs are under PntP1-GAL4/Imp-RNAi control (Figure 4A). Can the authors provide an explanation or data that Imp levels are reduced in quiescent NBs in Imp RNAi conditions compared to the wild-type at 24-48 ALH, and also discuss why Imp RNAi is active despite the lack of GFP expression? Is it possible that other cell types (e.g. glia) might be affected by the driver?

Thank you for the observation and we can see that this point was not made clear in the text. We have added the following text to the methods: "We were unable to quantify Imp fluorescence in quiescent TIINBs in ImpRNAi flies because quiescent TIINBs cannot be distinguished from quiescent Type I neuroblasts."

3. Are all PntP1>GFP-positive NBs CycE-positive? The population of GFP-positive but CycE-negative cells may suggest that quiescence exit is a multi-step process where Imp plays a role in the initial step.

We are sorry that this was not made clear, the following change has been made to the text on page 4. "…proliferative neuroblasts in interphase are GFP+Dpn+CycE+ whereas quiescent neuroblasts are GFP-Dpn+CycE- (15,16)."

Minor comments

1. Figure 2A-B: Representative wild-type series of Type II NB images should be provided to match the quantification shown in Figure 2C.

We show the requested wild type stains in Figure 1B-C.

2. Line 213: Please provide replicate information in all figure legends. In particular, what does the symbol represent in Figure 4A and 4C? It should be clear how many brains were quantified per biological replicate.

We changed the symbol “#” to “Number” – i.e. the Y axis shows the number of proliferating type II neuroblasts (of which there is a maximum number of 16 per brain). Replicate information has been added to all figure legends.

3. Figure 4E-F: Syed et al., 2017 (doi.org/10.7554/eLife.26287) have shown lack of Syp expression in Type II NBs at stages before 48 ALH and upon disruption of ecysone signaling. Can the authors explain the discrepancy?

Thank you for this observation. We have made the following change to the results section: “As expected, we found Syp to be expressed at lower levels than Imp in both proliferating and quiescent TIINBs (Figure 4E, third and fourth columns; quantified in 4F). Previous work has shown little to no Syp expression in early TIINBs; the very low levels of Syp seen here may be due to more sensitive acquisition methods than used previously (Syed et al., 2017)”.

4. Line 197: Immunofluorescence quantification method should be explained in more detail. Was the Raw Integrated Density calculated over 3D volume of the NBs or on select 2D planes? Why were nuclear areas and intensities removed from the analysis? Were any background subtraction method used?

We agree, and have made the following changes to the methods: "In FIJI, TIINBs were manually selected in a 2D plane at the largest cross section of the TIINB with the polygon lasso tool, and the area and Raw Integrated Density (RID) was measured. Imp is cytoplasmic and measuring fluorescence in the nucleus functioned as background subtraction."

5. In Figure 2C, how were the fluorescence intensity signals normalised between biological replicates and different genotypes?

Thank you for this comment. We have made this addition in the methods: "Imp is cytoplasmic and measuring fluorescence in the nucleus functioned as background subtraction."

6. Line 168: The role of Syp in entering the embryo-to-larval neuroblast quiescence is not yet established. The text should be revised. Perhaps the authors meant 'decomissioning and cell cycle exit'? 

Thanks for catching this error. We have changed the relevant sentence from the discussion. "Interestingly, Syp levels in quiescent TIINBs were higher than Syp levels in proliferative TIINBs, showing a correlation between high Syp levels and neuroblast quiescence, and consistent with earlier work showing Syp is required to elevate levels of nuclear Prospero and initiate decommissioning (21)."

---

## [Editor Report · Decision Letter 1]

7 Sep 2022

Imp is required for timely exit from quiescence in Drosophila type II neuroblasts

PONE-D-22-19798R1

Dear Dr. Doe,

We’re pleased to inform you that your manuscript has been judged scientifically suitable for publication and will be formally accepted for publication once it meets all outstanding technical requirements.

Kind regards,

Hongyan Wang, Ph.D.

Academic Editor

PLOS ONE

---

## [Editor Report · Acceptance letter]

9 Sep 2022

PONE-D-22-19798R1 

Imp is required for timely exit from quiescence in *Drosophila* type II neuroblasts 

Dear Dr. Doe:

I'm pleased to inform you that your manuscript has been deemed suitable for publication in PLOS ONE. Congratulations! Your manuscript is now with our production department. 

Kind regards, 

on behalf of

Dr. Hongyan Wang 

Academic Editor

PLOS ONE